# Presence of *Rickettsia* Species in Ticks Collected from Companion Animals in Northeastern Georgia, United States

**DOI:** 10.3390/vetsci8030037

**Published:** 2021-02-26

**Authors:** Hannah Stanley, DeLacy V. L. Rhodes

**Affiliations:** Department of Mathematics and Natural Sciences, Berry College, Mount Berry, GA 30149, USA; Hannah.Stanley@vikings.berry.edu

**Keywords:** companion animals, *Rickettsia*, tick-borne diseases, tick prevention

## Abstract

Tick-borne diseases are a major threat to both humans and their pets; therefore, it is important to evaluate the prevalence of pathogens carried by ticks on companion animals. In this study, attached and unattached Ixodid ticks were removed from companion animals by a veterinary practice in Hall County, Georgia. DNA was extracted from unengorged adult ticks and each was screened for the presence of *Rickettsia* spp. by polymerase chain reaction (PCR) and sequenced to determine the species present. Two hundred and four adult hard-bodied ticks were identified to species and *Rickettsia* spp. were found in 19.6% (*n* = 38) of the 194 analyzed DNA extracts. *Rickettsia montanensis* was found in *Dermacentor variablis* (14.7%; *n* = 25), *Amblyomma maculatum* (33.3%; *n* = 2), and *Rhipicephalus sanguineus* s.l. ticks (25%; *n* = 4). One *Amblyomma americanum* tick contained *Rickettsia amblyommatis*, while *Rickettsia felis* was found in one *Dermacentor variablis* tick, serving as the first report of *Rickettsia felis* in a tick in this region and within this tick vector. This study suggests that there is a risk of companion animals contracting a species of *Rickettsia* from a tick bite in northeastern Georgia, indicating a need for more investigation and highlighting the importance of tick prevention on pets.

## 1. Introduction

Tick-borne diseases are among the most prevalent vector-borne diseases in the United States. According to the Centers for Disease Control and Prevention, incidence rates for the most common tick-borne diseases in the United States, such as Lyme disease and Rocky Mountain spotted fever, have been steadily increasing over time. [1,2,3]. Although improved diagnostics and public awareness have increased understanding of the incidence and distribution of these tick-borne diseases, these infections still remain underrepresented due to lack of reporting and misdiagnosis. [4,5]. As cases of these tick-borne diseases continue to rise, the study of their causative agents is an essential step toward understanding their prevalence and decreasing their incidence.

Of the seven species of Ixodid ticks that bite and transmit disease to humans in the United States, five are found in Georgia, including *Dermacentor variabilis*, *Ixodes scapularis, Rhipicephalus sanguineus* (sensu lato), *Amblyomma maculatum,* and *Amblyomma americanum*. Each of these tick species is known to be the vector for different tick-borne infectious agents, including several different species of *Rickettsia* [6,7,8,9,10]. *Rickettsia* are small, Gram-negative intracellular bacteria that are broken into two traditional groups—the typhus group and the spotted fever group [11]. Members of the spotted fever group (SFG) *Rickettsia* cause several different illnesses such as *Rickettsia parkeri* rickettsiosis, caused by *R. parkeri*, and Rocky Mountain spotted fever (RMSF), caused by *R. rickettsii* [12,13]. An additional species of SFG *Rickettsia* more commonly associated with fleas, *R. felis*, has recently been found to be an important human pathogen in sub-Saharan Africa, where it is responsible for flea-borne spotted fever [14,15]. In addition to the species of *Rickettsia* that are associated with infectious disease, there are many species that are considered nonpathogenic. These various rickettsial endosymbionts, such as *R. bellii,* are associated with many different species of ticks and are not currently known to cause human illness [16].

Humans typically contract tick-borne pathogens through the bites of questing ticks that are acquired through outdoor activities or through close contact with animals transiently harboring ticks that are carrying pathogens. Companion animals can act as reservoirs for some tick-borne pathogens and are risk factors for all human age groups due to the introduction of ticks into people’s homes [17,18,19]. In addition to acting as hosts for infected ticks, domestic cats and dogs are susceptible to infection from certain tick-borne pathogens as well [20]. It is, therefore, important to evaluate the prevalence of pathogens carried by ticks on companion animals. In this study, ticks removed from companion animals by a veterinary practice in northeastern Georgia were screened individually for rickettsial species. By assessing the pathogen infection rates of ticks collected from companion animals, insight can be gained into the risk of humans and animals contracting these illnesses as well as into the risk companion animals may pose to their owners through the exposure of ticks that harbor these pathogens.

## 2. Materials and Methods

### 2.1. Tick Collection and Identification

Ticks used in this study were removed from companion animals between April and October of 2016 by a veterinarian office in Hall County, Georgia and stored in isopropyl alcohol. Attached and unattached ticks were removed during routine visits with 1–3 ticks removed per animal. For this study, only unengorged adult Ixodid ticks were analyzed. Ticks were identified visually using a dissecting microscope and a published identification guide [21].

### 2.2. DNA Extraction

DNA from all ticks was extracted using a commercially available DNA purification kit (GeneJET Genomic DNA Purification Kit, Thermo Fisher Scientific, Inc., Waltham, MA, USA). Instructions for extracting DNA from Gram-negative bacteria provided by the manufacturer were followed with minor alterations. An overnight digestion step was added before the manufacturer’s protocol steps were begun. Each tick was quartered and digested overnight with Proteinase K and included lysis buffer in a 56 °C water bath. Following ~16 h of digestion, the samples were thoroughly vortexed and centrifuged at 8000× *g* to pellet the tick exoskeleton. The supernatant was then transferred to a new tube and the manufacturer’s protocol was followed. Extracted DNA was stored in a refrigerator until screening and then frozen after.

### 2.3. DNA Amplification

DNA samples were screened individually by polymerase chain reaction (PCR) for members of the genus *Rickettsia*. Additionally, a control reaction was performed with each sample in which the tick 16S rRNA gene was amplified. The purpose of this control was to demonstrate that DNA was successfully extracted from each sample. The primers for this control were designed for this study (Tick 16 s Fwd: TTG CTG TGG TAT TTT GAC TAT ACA AAG GTA; Tick 16s Rev: CCG GTC TGA ACT CAG ATC). A commercially available Taq polymerase master mix (GoTaq^®^ Green Master Mix, Promega Biosciences, LLC, Madison, WI, USA) and its accompanying protocol were used for all screening reactions. Nested PCR was used for *Rickettsia* spp. detection by amplification of the *ompA* gene as previously described [22]. For use as a positive control, *ompA* was amplified from *R. parkeri* genomic DNA and cloned into pCR2.1 following the manufacturer’s instructions for TOPO cloning (Invitrogen, Carelsbad, CA). A negative control using water in the place of DNA was also included in all primary and secondary reactions. PCR products were run on 1% agarose gels (Bio-Rad, Hercules, CA, USA) and were stained with 1% ethidium bromide (Sigma-Aldrich, Saint Louis, MO, USA).

### 2.4. DNA Sequencing and Analysis

DNA sequencing was used to determine the species of all positive *Rickettsia* spp. samples [22]. The samples that yielded a positive *ompA* amplicon in the secondary reaction were purified from agarose gels using a gel extraction kit (QIAquick Gel Extraction Kit, Qiagen, Inc., Hilden, Germany) following the manufacturer’s instructions. Samples were sequenced by GenScript, LLC. (Piscataway, NJ, USA) on a fee for hire basis. Positive *ompA* sequences were analyzed using ClustalX for alignment [23].

The species of *Rickettsia* found in each tick was determined by BLASTn searching of the NCBI database and phylogenetic analysis using maximum likelihood methods [24]. For each sample, 544 base pairs of sequence were aligned and analyzed. Consensus sequences for comparison were acquired from GenBank and are listed in Appendix A along with accession numbers. Appendix A shows the percent similarities of all sequences analyzed in this study as determined by NCBI BLASTn searching, along with the sex of each individual tick that tested positive for *Rickettsia*. All sequences generated in this study are included in Appendix A and the phylogenetic tree is included as Appendix A.

## 3. Results

### 3.1. Tick Assemblages

A total of 204 adult unengorged ticks were collected for this study. Of these ticks, four species were found. A total of 180 were identified as *D. variabilis*, 16 as *Rhipicephalus sanguineus* s.l., six as *A. maculatum*, and two as *A. americanum* (Table 1). Of the collected ticks, 114 were male while 80 were female. No *I. scapularis* ticks were collected in this study, although this species of tick is present in Georgia.

### 3.2. PCR Screening and Species Identification

A total of 10 ticks tested negative with the tick 16S primer set, suggesting unsuccessful DNA extraction, and were removed from the analysis. This brought the total number of ticks tested for pathogen presence to 194. Of the 194 ticks tested, *Rickettsia* spp. were detected in 19.6% of all the ticks analyzed (*n* = 38). By BLASTn analysis and phylogenetic analysis, 31 of the positive samples were identified to be *R. montanensis*, 1 was *R. amblyommatis*, and 1 was *R. felis*. The one *R. amblyommatis* sequence was identified in an *A. americanum* tick while the one *R. felis* sequence was identified in a *D. variabilis* tick. The sequences that aligned with *R. montanensis* were found in 25 *D. variabilis* ticks (14.7%), 2 *A. maculatum* ticks (33.3%), and 4 *Rhipicephalus sanguineus* s.l. ticks (25%) (Table 1). Percentages in columns represent the percentage of each tick species positive for each *Rickettsia* species. Phylogenetic analysis supported BLAST sequence identification with high bootstrap values for each species identified (Appendix A). Five samples (Am002, Am003, Dv147, Dv162, and Dv127) tested positive for the presence of *Rickettsia* DNA by PCR but did not align with any tested *Rickettsia* species within the 98% sequence identity threshold values of this study. Therefore, these samples were removed from our analysis. Due to the sensitivity of nested PCR and the use of *R. parkeri* DNA as a positive control, this species of *Rickettsia* was omitted from this analysis.

## 4. Discussion

The most commonly collected ticks identified in this study were *D. variabilis* ticks, comprising 88% of the total ticks collected. Despite being the most common tick species found in Georgia, there were only two *A. americanum* ticks in the pool [5,9]. Additionally, it is interesting that although *Rhipicephalus sanguineus* s.l. prefers domestic dogs as hosts, only a small number of this species of tick (*n* = 16) was found on companion animals as part of this study [25,26]. As mentioned above, no *I. scapularis* were collected and analyzed in this study. This is most likely due to the fact that tick collections took place outside of the normal questing period for adult *I. scapularis* ticks in the southeastern United States [27]. A larger number of male ticks were analyzed than female ticks in this study. This is unsurprising due to the fact that only unengorged ticks were analyzed.

Approximately 19.6% (*n* = 38) of the ticks in this study harbored a species of *Rickettsia* with three different species of *Rickettsia* identified. Two known rickettsial species were found in one tick each—*R. felis* and *R. amblyommatis* (Table 1). The discovery of *R. felis* is an important and surprising finding. While the occurrence of infections in the United States is low, *R. felis* is a common cause of flea-borne fever worldwide [15,28]. This species was once known only as a flea-borne species, but it has since been identified that over 40 species of fleas, ticks, mites, and mosquitoes can harbor the pathogen [14,15]. Although the full vector competency of *R. felis* is still under investigation, several species of ticks have been found to harbor *R. felis* including *Haemophysalis suldata*, *Haemophysalis flava*, *Haemophysalis kitaokai*, *Ixodes ovata*, and *Rhiphicephalis sanguinius* [29,30,31]. Fewer studies have investigated *R. felis* in ticks in the United States, focusing instead on the prevalence of this pathogen in fleas and vertebrate hosts, though one US-based study did identify *R. felis* associated with *A. maculatum* ticks obtained from humans in the southern United States [32]. While co-feeding cannot be excluded in this study as a potential cause of the positive identification of *R. felis* in a tick, studies that have identified *R. felis* with environmentally or human-captured ticks along with the use of a tick cell culture model suggest that ticks may be able to harbor this pathogen [31,32,33]. This study identified one *D. variabilis* tick to be positive with *R. felis*, representing the first finding of *R. felis* in a tick in Georgia and the first finding of *R. felis* within a *Dermacentor* tick.

This study identified one *A. americanum* tick containing *R. amblyommatis*, a rickettsial species that is still being investigated to understand its full pathogenic potential. The investigation of humans seropositive for various species of SFG *Rickettsia* has suggested that *R. amblyomatis* (formerly known as *Candidatus* Rickettsia amblyomii) may be responsible for less severe cases of RMSF in the southeastern United States [34,35,36,37]. Another study looking into natural infection of dogs by ticks showed that dogs produced high antibody titers against *R. amblyommatis*, suggesting that dogs are able to be infected with this species [38]. As rickettsioses can be difficult to diagnose in canines, it is reasonable that this species may be a cause of disease in canine that is often overlooked [39].

The most prevalent species of *Rickettsia* identified in this study was *R. montanensis*, found in 34.1% (*n* = 31) of screened ticks. The ability of this rickettsial species to infect dogs has been investigated and it is not associated with any clinical symptoms in canines when infected both experimentally and naturally [38,40]. Additionally, it has been found that dogs mount a strong antibody response against *R. montanensis* and, though antibody cross-reactivity is common amongst the SFG rickettsiae, this antibody response does not provide protection against infection with some pathogenic species of *Rickettsia* [38,39,40]. In this study, twenty-five *D. variabilis* ticks (14.7%), two *A. maculatum* ticks (33.3%), and four *Rhipicephalus sanguineus* ticks (25%) were found to harbor *R. montanensis* (Table 1). Though nonpathogenic in dogs, there has been a single case of an afebrile rash illness in a six-year-old girl from Georgia who was bitten by a *D. variabilis* tick carrying *R. montanensis,* indicating that this organism may occasionally be able to cause disease and underscoring the importance of the use of tick preventative treatment on household pets [41]. Additionally, it is speculated that colonization by *R. montanensis* in the tick can outcompete *R. rickettsii* [7]. As this is a disease known to occur in northern Georgia, the prevalence of *R. montanensis* in ticks found in this study may have contributed to the absence of *R. rickettsii* in our samples [7].

This study is small in scope and contains limitations that are important to note. For one, the ticks collected in this study were donated by a veterinary clinic and no data were collected that associate the individual ticks with the animal(s) from whom they were collected. Therefore, information concerning the species of the associated animal hosts (i.e., cats versus dogs) and the number of ticks collected from each individual animal is unavailable. Additionally, one gene was assessed to determine *Rickettsia* species, though it was analyzed through multiple methods. Greater information into the individual ticks and the analysis of an additional gene for the determination of *Rickettsia* species would have strengthened our results.

## 5. Conclusions

Ticks collected from companion animals and veterinary offices provide a unique perspective into the human and animal risk for exposure to ticks and tick-borne diseases. Studying ticks collected in this manner provides important insight into the prevalence of tick-borne pathogens while also underscoring the need for regular tick prevention and removal from companion animals. This study has shown that pets can bring ticks potentially harboring pathogens into homes, increasing the risk of tick-borne diseases to pet owners. Almost a quarter of all the ticks analyzed in this study (19.6%) contained some species of *Rickettsia* (Table 1). This study has also identified a known pathogenic species, *R. felis*, to be present in *D. variablis* ticks in Georgia. While greater information into the history of the ticks and animals utilized in this study and additional genetic analysis could provide important information to the analysis of this study, this work has regional significance and provides data that are important in helping veterinarians impress the need for regular tick prevention on pets.

## Figures and Tables

**Table 1 vetsci-08-00037-t001:** Total numbers of tick species and associated *Rickettsia* species.

***Rickettsia* Species**		**Tick Species**				**Total Per Species (% Positive)**
		***Dermacentor variablis***	***Amblyomma americanum***	***Amblyomma maculatum***	***Rhipicephalus sanguineus* s.l.**	
	*R. montanensis*	25 (14.7%)	0 (0%)	2 (33.3%)	4 (25%)	31 (34.1%)
	*R. felis*	1 (0.6%)	0 (0%)	0 (0%)	0 (0%)	1 (1.01%)
	*R. amblyommatis*	0 (0%)	1 (50%)	0 (0%)	0 (0%)	1 (1.01%)
	Unknown	3 (1.8%)	0 (0%)	2 (33.3%)	0 (0%)	5 (5.5%)
Total *Rickettsia* Positive/Total Tick Species (% positive)		29/170 (17%)	1/2 (50%)	4/6 (66.7%)	4/16 (25%)	38/194 (19.6%)

## Data Availability

All data generated in this study can be found in Appendix A.

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
