# Peer review of "Presence of Rickettsia Species in Ticks Collected from Companion Animals in Northeastern Georgia, United States"

_vetsci, 2021, doi:10.3390/vetsci8030037_

Round 1

Reviewer 1 Report

Unfortunately, I do not think the corrections presented compensated the initial flaws in the experimental design and the molecular ID of the Rickettsia species.

Author Response

Dear Reviewer 1,

Thank for for taking the time to review my paper. I appreciate your efforts and your constructive criticisms. Please see the attached cover letter for a response to each reviewers' comments. 

Sincerely,

DeLacy V. L. Rhodes

Reviewer 2 Report

The authors have improved their manuscript and I recommend it be published.

There were a few minor mistakes that can be easily corrected:

Review of revised ms dealing with ticks and rickettsia from NE Georgia.

Line 67: change ticks to tick. I.e. DNA from each tick(s) was extracted . . .

Line 108: add an “s” to tick, I.e. Of the collected tick(s), 114 were . . .

Line 160: use italics with Rickettsia and reword sentence, I.e. . . . seropositive humans for various SFG Rickettsia species have suggested . . .

Author Response

Dear Reviewer 2,

Thank you for your time and efforts in reviewing my manuscript. I have incorporated your suggested changes. Please see the attached cover letter for a response to each reviewers' comments.

Sincerely,

DeLacy V. L. Rhodes

Reviewer 3 Report

Thank you for answering to my previous questions and amend your manuscript. I think it is ready for publication.

Line 107-108:  A. maculatum should be in italic.

Author Response

Dear Reviewer 3,

Thank you for your time and efforts in the review of my manuscript. I have incorporated the change you suggested. Please see the attached cover letter for a detailed response to each reviewer comment.

Sincerely,

DeLacy V. L. Rhodes

This manuscript is a resubmission of an earlier submission. The following is a list of the peer review reports and author responses from that submission.

Round 1

Reviewer 1 Report

This is a nice study and despite being a clinician, I could follow it easily. I would only advise the authors to provide the results regarding ticks and Rickettsia species distinguishing those found in dogs and those in cats. This information could be added in the results section and also in the table 1. 

Author Response

Dear Reviewer 1,

Thank you for your time and consideration of our manuscript. We especially appreciate your comments on the readability of our manuscript. Please see attached our letter with a breakdown of your and the other reviewers' individual comments.

Sincerely,

DeLacy V. L. Rhodes

Reviewer 2 Report

Please see my comments below:

Below are some questions and suggested changes.

Abstract

Line 11: For attached ticks how are you assured that ticks were unengorged? But you assume they fed, so what is the difference from blood fed and unengorged?

Lines 14-17: The first time writing a name of a tick or rickettsia, you should write out the genus name.

Introduction

Line 36: gram should be capitalized, i.e. . . . Gram negative . . .

Line 37: Rickettsia species can be placed in to 3, 4, 5 and 13 groups depending upon which genetic typing you utilized. So, I would recommend adding the word “traditional” to line 37, i.e. . . . into two traditional groups - the typhus group and . . .

Line 44: Rickettsia monacensis is considered a human pathogen. 

Materials and Methods

Line 64: Pleased indicate whether ticks were processed and assessed individually or in pools.

Line 66: DNA can be singular or plural. In this case I assume it is plural so “was” should be “were”.

Line 70: kit’s should be kits.

Line 76: should name the PCR assay used to screen tick DNA samples for rickettsial DNA.

Line 85: add the manufacturer’s name, City, ST

Line 87: manufactures, Cities and STs for agarose and EtBr.

Results

Line 107: Rewrite this sentence (e.g. Though I. scapularis, normally/commonly/rarely found in Georgia, none were collected during this study.)

Line 116: Use italics for Rickettsia

Lines 118 and 119: Use italics for the three species of Rickettsia listed.

Figure 1: I don’t find this figure very informative. It could be deleted

Discussion

Line 154: Please write out the species of Haemophaslis, i.e. H. suldata, H. flava, and H. kitaokai.

Line 161: Use italics for D. variabilis.

Line 165: SFR should be SFG and use italics with Rickettsia.

Line 166: amblyommi has three m’s.

Conclusions

Line 189: This is the first time that it is indicated that some (n=?) ticks were collected from veterinary offices.

References

Use italics for scientific names in the references.

Author Response

Dear Reviewer 2,

Thank you for your time and consideration of our manuscript. We especially appreciate your comments and close attention to detail. Please see attached our letter with a breakdown of your and the other reviewers' individual comments.

Sincerely,

DeLacy V. L. Rhodes

Reviewer 3 Report

The present work brings interesting information about the circulation of Rickettsia species in ticks collected from dogs and cats (?) in veterinary clinics in Georgia, southeastern USA. Even though the results are interesting, additional PCR assays targeting at least one more molecular marker should be performed (e.g. gltA, htrA). NJ does not provide an accurate phylogenetic positioning of the detected sequences. I strongly recommend authors to use Bayesian Inference or Maximum Likelihood methods. Also, authors should show the size of the alignment and the evolutionary model used for each target gene (again, at least two). In order to save some money, authors could pick some representative samples that showed to be positive for each of the Rickettsia species. I encourage authors also to show a concatenated phylogenetic tree with the two chosen target genes.

  • The phylogenetic analysis should be re-done. In fact, R. felis belongs to the translation group instead of SFG. Rickettsia species from the TG (Typhus group) should be used as an outgroup.
  • Which tick species were collected from dogs and cats respectively? 
  • Are the authors sure that the collected ticks were unengorged? I mean, this is easily observed for female ticks, but considering that male ticks ingested a lower amount of blood compared to females, it is quite difficult to state the males did not ingest a blood meal. Besides that, male hard ticks usually feed on the vertebrate hosts BEFORE females, in order to signalize them the best stop to get a blood meal. I strongly recoommend authors be careful about saying they only sampled unengorged ticks. Besides that, the positive ticks were attached or non-attached to the host skin?
  • When it comes to co-feeding and R. felis transmission: did the authors found co-infestation by fleas in the dog that they recovered the R. felis-tick? 
  • Authors should italicize all scientific names throughout the manuscript.
  • Line 154: R. sanguineus
  • Last but not least: authors should include the UNKNOWN Rickettsia species in the phylogenetic tree. This is an interesting result that would bring more attention the MS. What was the highest hit in the BLAST analysis? Again, authors should run additional PCR assays targeting more molecular markers in order to phylogenetically position this uncharacterized Rickettsia sequences.

Author Response

Dear Reviewer 2,

Thank you for your time and consideration of our manuscript. We especially appreciate your comments and suggestions for strengthening our findings. Please see attached our letter with a breakdown of your and the other reviewers' individual comments. 

Sincerely,

DeLacy V. L. Rhodes

Round 2

Reviewer 3 Report

Unfortunately and based on the response letter of authors, I do not think the presented manuscript is scientifically sound. Therefore, I do not recommend the publication of the present article without the modifications proposed earlier.